# One-Stage Synovectomies Result in Improved Short-Term Outcomes Compared to Two-Stage Synovectomies of Diffuse-Type Tenosynovial Giant Cell Tumor (D-TGCT) of the Knee: A Multicenter, Retrospective, Cohort Study

**DOI:** 10.3390/cancers15030941

**Published:** 2023-02-02

**Authors:** Geert Spierenburg, Floortje G. M. Verspoor, Jay S. Wunder, Anthony M. Griffin, Peter C. Ferguson, Matthew T. Houdek, David M. King, Richard Boyle, Robert Lor Randall, Steven W. Thorpe, Jacob I. Priester, Erik J. Geiger, Lizz van der Heijden, Nicholas M. Bernthal, Bart H. W. B. Schreuder, Hans Gelderblom, Michiel A. J. van de Sande

**Affiliations:** 1Department of Orthopedic Surgery, Leiden University Medical Center, 2333 ZA Leiden, The Netherlands; 2Department of Orthopedic Surgery, Amsterdam University Medical Centers, 1105 AZ Amsterdam, The Netherlands; 3Division of Orthopaedic Surgery, Mount Sinai Hospital, Toronto, ON M5G 1X5, Canada; 4Department of Orthopaedic Surgery, Mayo Clinic, Rochester, MN 55905, USA; 5Department of Orthopaedic Surgery, Medical College of Wisconsin, Milwaukee, WI 53226, USA; 6Department of Orthopedic Surgery, Royal Prince Alfred Hospital, Sydney, NSW 2050, Australia; 7Department of Orthopaedic Surgery, University of California-Davis, Sacramento, CA 95817, USA; 8Rothman Institute and Department of Orthopedic Surgery Thomas Jefferson University, Philadelphia, PA 19107, USA; 9Department of Orthopaedic Surgery, University of California-Los Angeles, Los Angeles, CA 90404, USA; 10Department of Orthopaedics, Radboudumc, 6525 GA Nijmegen, The Netherlands; 11Department of Medical Oncology, Leiden University Medical Center, 2333 ZA Leiden, The Netherlands

**Keywords:** tenosynovial giant cell tumor, TGCT, diffuse-type, knee, synovectomy, one-stage, two-stage

## Abstract

**Simple Summary:**

Diffuse-type tenosynovial giant cell tumor (D-TGCT) is a rare disease that can be located on the knee joints’anterior and posterior sides. Surgery approaching both sides of the knee is often performed to remove the tumor. However, there is no consensus on whether surgery on both sides should be performed during one procedure or in two separate interventions. In this retrospective, cohort study, 191 patients were included from nine sarcoma centers worldwide. The goal was to compare the short-term postoperative outcomes of two-sided surgeries for D-TGCTs of the knee, performed in either one or two stages. Surgery on the knee’s anterior and posterior sides performed in one stage did not result in impaired rehabilitation compared to two-stage surgery. Additionally, patients undergoing surgery in one stage had a shorter hospital stay and no more complications.

**Abstract:**

Diffuse-type tenosynovial giant cell tumors’ (D-TGCTs) intra- and extra-articular expansion about the knee often necessitates an anterior and posterior surgical approach to facilitate an extensive synovectomy. There is no consensus on whether two-sided synovectomies should be performed in one or two stages. This retrospective study included 191 D-TGCT patients from nine sarcoma centers worldwide to compare the postoperative short-term outcomes between both treatments. Secondary outcomes were rates of radiological progression and subsequent treatments. Between 2000 and 2020, 117 patients underwent one-stage and 74 patients underwent two-stage synovectomies. The maximum range of motion achieved within one year postoperatively was similar (flexion 123–120°, *p* = 0.109; extension 0°, *p* = 0.093). Patients undergoing two-stage synovectomies stayed longer in the hospital (6 vs. 4 days, *p* < 0.0001). Complications occurred more often after two-stage synovectomies, although this was not statistically different (36% vs. 24%, *p* = 0.095). Patients treated with two-stage synovectomies exhibited more radiological progression and required subsequent treatments more often than patients treated with one-stage synovectomies (52% vs. 37%, *p* = 0.036) (54% vs. 34%, *p* = 0.007). In conclusion, D-TGCT of the knee requiring two-side synovectomies should be treated by one-stage synovectomies if feasible, since patients achieve a similar range of motion, do not have more complications, but stay for a shorter time in the hospital.

## 1. Introduction

Tenosynovial giant cell tumors (TGCTs) are typically monoarticular diseases, emerging from the synovial lining of joints, bursae, and tendon sheaths [1]. The tumor is composed of neoplastic and reactive components, both driven by CSF1 overexpression [2]. TGCTs comprise two main subtypes: localized-type (L-TGCTs) and diffuse-type TGCTs (D-TGCTs). Both subtypes are histologically identical and are distinguished by their differing radiological pattern and clinical behavior [1]. Malignant TGCTs are considered a third subtype; however, this is only incidentally reported [3].

D-TGCTs behave locally more aggressively, and disease control is more challenging compared to L-TGCTs [1,4,5,6]. This study focuses on patients with D-TGCTs. The incidence rate of D-TGCTs is estimated to be 5 to 8 per million person years, and has its onset in a relatively young population, mostly between 30 and 50 years of age [7,8]. D-TGCTs affect large joints, in particular the knee. Common symptoms are pain, swelling, stiffness, and limited function; therefore, D-TGCTs can significantly impair patients’ quality of life [9,10]. These unspecific symptoms often lead to diagnostic delays [11]. Diagnosis is made through MRI and histological confirmation. D-TGCTs are characterized by a multilobulated lesion (>5 cm) with indistinct borders on MRI, and can be located both intra- and extra-articularly [12]. Additionally, its locally aggressive behavior can result in joint deterioration caused by inflammatory conditions and infiltrative growth. 

To date, surgery is regarded as the backbone of treatment to relieve symptoms and prevent joint deterioration [13]. Surgery by means of synovectomy aims to remove all tumors macroscopically to increase the chance of favorable outcomes [14,15]. However, achieving complete resection may result in iatrogenic morbidity if neurovascular structures are involved or because D-TGCTs’ extensive growth necessitates large incisions and surgical exposures. Synovectomies for D-TGCTs are associated with recurrence free-survival of 40% at 10 years [5,15,16]. The elucidation of the CSF1R driver mechanism led to the use of new therapeutic modalities, such as CSF1R inhibitors [17,18,19]. CSF1R inhibitors are indicated for patients not amenable to surgery but have only limited availability to date. While the US Food and Drug Administration approved one CSF1R inhibitor, pexidartinib (Daiichi Sankyo, Tokyo, Japan), for D-TGCTs, the European Medicines Agency and Health Canada declined market authorization due to an unfavorable risk–benefit ratio [20]. Therefore, extensive synovectomies still remain a mainstay of treatment. Nevertheless, a consensus regarding the optimal surgical approach has not been reached [13]. A recent meta-analysis by Chandra et al. estimated a 1.56 increased risk of recurrence after arthroscopic surgical management of D-TGCTs of the knee compared to an open approach [21]. Furthermore, D-TGCTs in the knee often requires incisions from the anterior and posterior sides to remove all intra- and extra-articular diseases. It remains undecided whether operating on the anterior and posterior sides should be performed in one or two stages [22,23,24]. One-stage synovectomies are arguably less invasive for patients, as undergoing only one surgery requires one recovery period. Hypothetically, a one-stage synovectomy could result in impaired postoperative recovery and increased complications risk with simultaneous wounds on two sides of the knee. This study aims to compare the short-term outcomes of one- versus two-stage synovectomies of the anterior and posterior sides performed for D-TGCTs of the knee. A multicenter collaboration was initiated to bundle the experiences and data of several sarcoma centers worldwide. 

## 2. Materials and Methods

In this international, multicenter, retrospective observational cohort study, patients that had a synovectomy of the anterior and posterior side of the knee for D-TGCTs between January 2000 and June 2021 were eligible. All consecutive patients were included from nine specialized sarcoma centers in the Netherlands, United States, Australia, and Canada.

All patients had histologically confirmed TGCTs located in the knee. Additionally, they underwent a two-sided synovectomy of the knee performed in one or two stages. In a one-stage synovectomy, the anterior and posterior sides of the knee were operated on during the same surgery. A two-stage synovectomy was defined as two separate surgeries, one addressing the anterior side and the other the posterior side. The separate surgeries must have been performed within six months to be defined as a two-stage synovectomy. The order of approach (i.e., first anterior or first posterior) or the surgical technique (open or arthroscopic) were not exclusion criteria. 

### 2.1. Two-Sided Synovectomy for D-TGCTs of the Knee

D-TGCTs in the knee are often located throughout the joint due to their multicompartmental growth pattern [12]. Common locations on the anterior side are the patellar recesses, the medial and lateral gutter, Hoffa’s fat pad, and the anterior cruciate ligament. Posterior, D-TGCTs are typically located beneath the gastrocnemius insertions and intercondylar recesses, around the posterior cruciate ligament, and in the Baker’s cyst around the hamstring tendons. Extra-articular locations often occur with extensive intra-articular growth. 

Total synovectomy of the ventral side comprises removal of the synovium, often including the entire capsule and the suprapatellar bursa (Figure 1A). In addition, all tumor around the patella, along the femur, Hoffa’s fat pad, in the posterolateral and posteromedial spaces, and surrounding the anterior cruciate ligament should be removed (Figure 1B). Bone erosions are often located in the notch, around the femoral origin of the medial collateral ligament and the posterior tibial plateau (Figure 1C,D). Parts of the posterior recesses can be accessed through the ventral approach, but a separate posterior exposure is commonly required. 

For the posterior approach, a lazy S-shaped incision is made before dissecting the popliteal fascia (Figure 1E). Commonly, the involvement of the hamstring tendons coincides with tumor located in a Baker’s cyst (Figure 1F,G). After deeper dissection and retraction of the gastrocnemius muscle, posterior tumor in the subgastrocnemius recess appears (Figure 1H). Additional necessary approaches can be made medial to the semimembranosus, between the semimembranosus and the popliteal vessel and tibial nerve, and between the popliteal vessels and peroneal nerve for tibial–fibular joint involvement. The popliteal artery, tibial, peroneal, and sural nerves, and the small saphenous vein are at risk during this approach.

During a one-stage synovectomy, patients are turned from a prone to a supine position or vice versa intraoperatively. 

### 2.2. Data

All data were retrospectively collected from patient medical records and pseudonymized before transferring to the principal investigators. The following data were collected: patient demographics, prior treatments, preoperative clinical presentation, date(s) and type(s) of surgical interventions, length of hospital stay counting from the day of surgery till the day of discharge, postoperative range of motion up to one year, the need of walking aids, surgery-related complications, radiological progression, and subsequent treatments. For two-stage synovectomies, the length of hospital stay and surgical duration of the two separate surgeries were added together. In addition, radiological progression was measured from the date of the second intervention to the date of progression for two-stage synovectomies to avoid immortal time bias. 

The primary aim of this study was to compare short-term outcomes between one- and two-stage synovectomies, such as surgical duration, length of hospital stay, postoperative range of motion within the first year after surgery, and complications. Secondary outcomes were radiological progression, clinical improvement, and the need for subsequent treatments. 

This study was performed according to the Declaration of Helsinki and was approved by the Institutional Review Board from the Leiden University Medical Center.

### 2.3. Statistical Analysis 

Continuous data were described by medians and ranges, and categorical data by the number of observations and percentages (%). Rates were calculated for the available data in individual categories. For all data, patients were stratified by undergoing a one- or two-stage synovectomy. Chi-square, Mann–Whitney U, or Kruskal–Wallis tests were performed to compare independent variables between the groups. 

Finally, we performed subgroup analyses comparing only open one-stage and two-stage synovectomies. Due to the low incidence of TGCTs, no formal sample size calculation was performed, and all patients that fulfilled the inclusion criteria were included. IBM Statistical Package for Social Statistics 25 (Chicago, IL, USA) was used for analysis. 

## 3. Results

Between January 2000 and June 2021, 191 consecutive patients underwent a one- or two-stage synovectomy of the anterior and posterior side of the knee for D-TGCTs. Of these 191 patients, 117 underwent a one-stage synovectomy and 74 a two-stage synovectomy. No significant differences were found between age, gender, and admission status (i.e., therapy naïve or prior treatment) between the two subgroups (Table 1). However, the participating sarcoma centers differed in their preferences for performing one- or two-stage synovectomies. Three sarcoma centers performed only one-stage synovectomies, one center performed only two-stage synovectomies, and both methods were used in the remaining centers (Table 1).

Of the 191 patients, 10 underwent a second one- or two-stage synovectomy, totaling 201 interventions. These interventions were comprised of 126 one-stage and 75 two-stage synovectomies. The preoperative range of motion, including a flexion of 120 degrees and no extension lag, was equal, and the surgeries in both groups were performed around the same period (Table 2). The one-stage synovectomies were performed either completely open, completely arthroscopic, or with both techniques combined. Conversely, most two-stage synovectomies were performed solely open, and a combined technique was only used in a few cases (*p* < 0.0001). The median interval between the first and second intervention of two-stage synovectomies was 2 months (range 0–6 months). The length of hospital stay was longer for patients undergoing a two-stage synovectomy (sum of two admissions) (*p* < 0.0001) (Table 2). Postoperative knee flexion motion measured across multiple time points postoperatively was equal at 3, 6 and 12 months postoperatively, and the maximum range of motion reached within the first year after treatment was not different between the two groups (Figure 2, Table 2). Complications occurred more often in the patients undergoing a two-stage synovectomy (*p* = 0.095), although this was not statistically significant. In both groups, superficial wound infections and wound healing problems were the most common complications. Three deep wound infections occurred after two-stage synovectomies. Six patients required walking aids at six months postoperatively, consisting of elbow crutches and canes. Four of these six patients underwent a one-stage synovectomy (2%), and the others a two-stage synovectomy (3%). At one year, only two patients still used a cane, one from each group. 

Median follow-up for patients undergoing a one-stage or two-stage synovectomy was 45 and 59 months, respectively (*p* = 0.047) (Table 3). The progression rate for patients undergoing a one-stage synovectomy was 37%, and the progression rate was 52% for the two-stage group (*p* = 0.036). However, this finding was no longer significant after performing a Kaplan–Meier analysis (log-rank test *p* = 0.080) (Figure 3). Additionally, patients undergoing a two-stage synovectomy required subsequent treatments significantly more often than after a one-stage synovectomy (54% vs. 34%; *p* = 0.007). Patients were mainly retreated by a repeat synovectomy (Table 3). 

At six months and one year postoperatively, pain and swelling were the symptoms that most frequently improved compared to the preoperative status. While stiffness commonly improved in patients undergoing one-stage synovectomies (34% at one year postoperatively), it only improved in three patients at six months (7%) and in one at one year (2%) of the patients undergoing a two-stage synovectomy (Figure 4). 

### Subgroup Analyses

After comparing only open one-stage and two-stage synovectomies, patients achieved a similar range of motion within the first year after surgery (flexion 125–120°, *p* = 0.126; extension 0°, *p* = 0.253), and the median length of hospital stay was equal (6 days); however, the distribution was significantly longer for patients undergoing two-stage synovectomies (*p* = 0.008) (Appendix A). Additionally, complications occurred more frequently following a two-stage procedure, although this finding was not significant (36% vs. 22%, *p* = 0.069) (Appendix A). 

## 4. Discussion

There is a need for new therapeutic modalities, which are in development; meanwhile, surgery remains the standard treatment for TGCTs. A complete (anterior and posterior) synovectomy is often indicated in diffuse-type TGCTs of the knee, but consensus regarding the ideal surgical procedure is lacking [13,25]. This is the first multicenter study with a large cohort comparing the short-term outcomes of one- and two-stage synovectomies of the anterior and posterior sides of the knee [23,24,25]. Patients undergoing one-stage synovectomies achieved an equal range of motion postoperatively, but stayed shorter in the hospital and had fewer complications. Thus, one-stage synovectomies are preferred over two-stage synovectomies if feasible. 

While most previous studies focused on different techniques used during two-sided synovectomies (open or arthroscopic), this study is the first to compare the effect of one or two stages [22,23,24,25]. Since simultaneous surgery on the anterior and posterior side of the knee is more invasive for patients, this can lead to discouraging surgeons from performing a one-stage synovectomy in some cases. Although prolonged rehabilitation may not be desirable in some cases (e.g., elderly patients), the typical population affected by D-TGCTs is relatively young, as also shown by this cohort and in accordance with the literature [7,8]. Younger patients can cope better with invasive procedures in general [26]. Patients undergoing one-stage synovectomies had an equal range of motion at 3, 6 and 12 months and achieved the same range of motion within the first year after surgery compared to patients undergoing two-stage synovectomies (median flexion 123 degrees with full extension). In conclusion, patients undergoing one-stage synovectomies do not have an impaired recovery and achieve the range of motion required to perform activities of daily living [27].

On the other hand, for two-stage synovectomies, the range of motion was measured after the second intervention, and the median interval of 2 months between the first and second surgery was not taken into account. In this period, patients are still recovering from the first intervention, resulting in prolonged rehabilitation for patients undergoing a two-stage synovectomy. Besides more extended rehabilitation, patients undergoing two-stage synovectomies also had to stay longer in the hospital due to two separate interventions. The length of admission was not affected by the invasiveness of one-stage synovectomies. Repeated admissions for surgery and longer lengths of hospital stay have a negative impact on TGCT-related medical costs [10]. Additionally, prolonged rehabilitation results in a longer return to work time and daily activities such as sports. 

Approaching the anterior and posterior sides of the knee surgically simultaneously did not result in higher complication rates following one-stage synovectomies. Contrariwise, two-stage synovectomies led to higher complication rates following two separate interventions, although not significantly. Compared to the study of Mastboom et al., the total complication rate in this cohort was relatively high (28% vs. 12%) [5]. Selection bias may have been introduced since only patients were included with D-TGCTs located on the anterior and posterior side of the knee, resulting in a cohort with more severe presentations and requiring more invasive surgeries.

Radiological progression, a secondary outcome, occurred more frequently after two-stage synovectomies, although this finding was not significant after a Kaplan–Meier analysis. In addition, patients undergoing a two-stage synovectomy required repeated treatments more often. However, local surveillance and treatment regimens may have biased these results, since local recurrences are not always symptomatic and do not require treatment in every case. 

## 5. Limitations

Since patients were not randomized to either one- or two-stage synovectomies, some biases may have been introduced. First, sarcoma centers significantly differed in performing one- or two-stage synovectomies. Some centers performed only one approach, which could introduce information bias toward this approach. Secondly, bias is possible by surgeons or patients who prefer one of the two interventions. Finally, indication bias might have been introduced by sarcoma centers performing both procedures. Complete arthroscopic approaches were only performed in one-stage synovectomies, which could suggest that this treatment group included less severe cases. When looked at in more detail, no considerable differences were found per center between the disease status at the time of surgery (primary or recurrent disease) for patients undergoing one- or two-stage synovectomies (Appendix A). Unfortunately, no more data were available to assess disease severity based on tumor size, tumor localization, or patient-reported outcome measurements. Nonetheless, results regarding the short-term outcomes were not different after analyzing subgroups of patients treated by solely open procedures. 

The authors agree that removing D-TGCTs arthroscopically is technically challenging and requires a lot of training to achieve complete tumor resection [28]. Thus, removing extensive D-TGCTs located anterior and posterior arthroscopically should only be performed by sarcoma centers with skilled arthroscopists.

Due to the retrospective multicenter study design, no standardized follow-up schemes were followed. This may have resulted in different rates of radiological progression and indications for subsequent treatments per center, since local recurrences can be asymptomatic and do not always necessitate treatment. 

Finally, no prospective data were collected due to the retrospective design. Therefore, no validated classification criteria could have been compared, such as tumor volume score used to measure radiological progression or patient-reported outcome measurements to quantify the health-related quality of life. 

A randomized controlled trial will minimize the risk of these biases, and validated measurements can be integrated. Although this would be the ideal study design to compare both approaches, performing a prospective trial for a rare disease such as D-TGCT is challenging but not impossible [11]. 

## 6. Conclusions

A synovectomy of the anterior and posterior sides of the knee is often required to remove all advanced D-TGCTs macroscopically. This retrospective multicenter study showed that one-stage synovectomies do not result in impaired rehabilitation compared to two-stage synovectomies. Additionally, patients undergoing a one-stage synovectomy had a shorter length of hospital stay and no more complications than patients undergoing two-stage synovectomies.

## Figures and Tables

**Figure 1 cancers-15-00941-f001:**
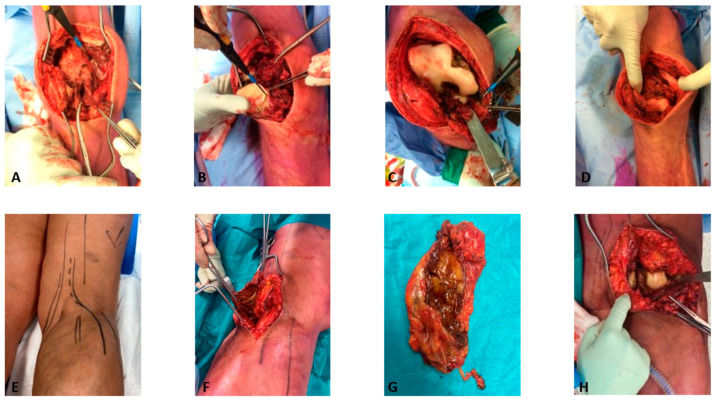
Two-sided synovectomy of the knee. An anterior synovectomy for D-TGCT about the knee is illustrated in figures (**A**–**D**). A posterior synovectomy for D-TGCT about the knee, in which also a Bakers’ cyst is also removed, is illustrated in figures (**E**–**H**).

**Figure 2 cancers-15-00941-f002:**
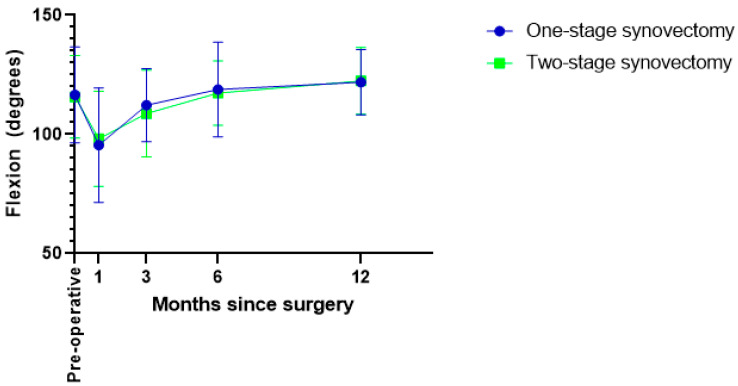
Course of the postoperative flexion range.

**Figure 3 cancers-15-00941-f003:**
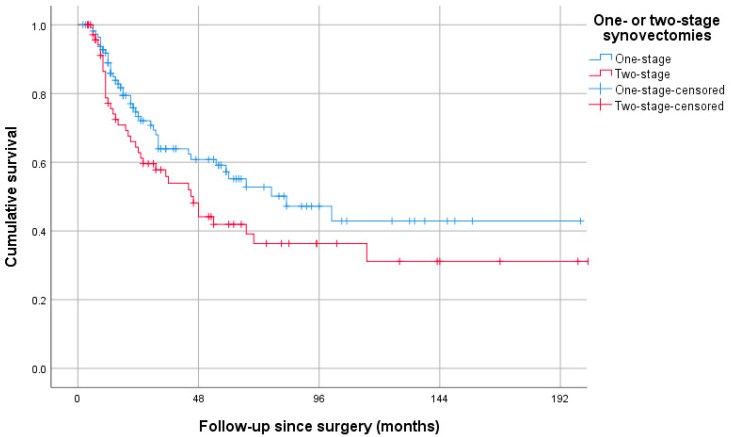
Postoperative progression-free survival; Kaplan–Meier analysis.

**Figure 4 cancers-15-00941-f004:**
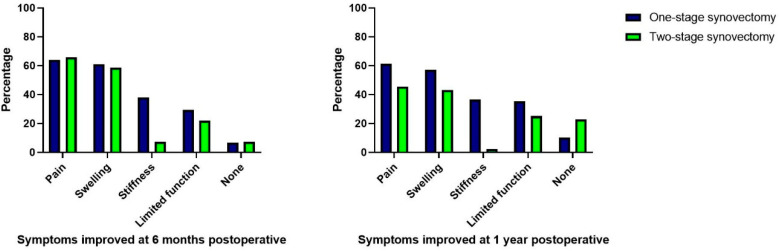
Symptoms improved at six months and one year postoperatively.

**Table 1 cancers-15-00941-t001:** D-TGCT patient baseline characteristics.

Features	One-Stage Synovectomy	Two-Stage Synovectomy	*p*-Value
	N = 117	N = 74	
Age, median (range)	39 (14–74)	37 (14–65)	0.717
Gender			0.46
Male	57	32	
Female	60	42	
Centers			<0.0001
LUMC	20	19	
RUMC	17	19	
MSH	31	-	
AUMC	6	17	
MAYO	11	6	
MCW	15	-	
RPAH	-	12	
UCD	10	-	
UCLA	7	1	
Prior treatments *	N = 113	N = 73	0.548
None	54	39	
Yes	59	34	
Synovectomy	56	33	
Systemic therapy	4	1	
RSO	2	4	
EBR	-	2	
Unknown	4	-	

* Sum of observations can be more than the total number of individual patients; RSO—Radiosynoviorthesis, EBR—External Beam Radiotherapy.

**Table 2 cancers-15-00941-t002:** Surgery characteristics of all interventions.

Features	One-Stage Synovectomy	Two-Stage Synovectomy	*p*-Value
	N = 126	N = 75	
Preoperative range of motion, degrees, median (range)	N = 108	N = 45	
Flexion			
Extension	120 (30–150)	120 (90–140)	0.63
	0 (0–20 ^†^)	0 (0–15 ^†^)	0.83
Median year of surgery (range)	2015 (2002–2021)	2013 (2002–2020)	0.02
Surgical technique	N = 123	N = 75	<0.0001
Open	58 (47%)	67 (89%)	
Combined ^a^	51 (42%)	8 (11%)	
Arthroscopic	14 (11%)	-	
Length of hospital stay, days,	N = 124	N = 71	<0.0001
Median (range) ^b^	4 (1–13)	6 (3–26)	
Maximum range of motion	N = 114	N = 49	
PO ^c^, degrees, median (range)			
Flexion	123 (75–145)	120 (95–140)	0.109
Extension	0 (0–30 ^†^)	0 (0–10 ^†^)	0.073
Complications *	N = 123	N = 72	0.095
Yes	29 (24%)	27 (36%)	
Wound healing problems	10	9	
Superficial wound infection	8	12	
Deep wound infection			
Joint stiffness	-	3	
Hemarthrosis	1	2	
Neurovascular damage	3	3	
Thrombosis	3	2	
Other	1	-	
	9	1	

^a^ Combined comprises arthroscopic synovectomy of the anterior side and open synovectomy of the posterior side; ^b^ for two-stage synovectomy, the sum of both surgeries is calculated; ^c^ PO—postoperative. ^†^ The number of degrees equals the degrees of extension lag; * sum of observations can be more than the total number of individual patients.

**Table 3 cancers-15-00941-t003:** Postoperative course.

Features	One-Stage Synovectomy	Two-Stage Synovectomy	*p*-Value
	N = 117	N = 74	
Median follow-up, months (range)	45 (1–200)	59 (3–203)	0.047
Radiological progression	N = 115	N = 73	
Yes	42 (37%)	38 (52%)	0.036
Months till radiological progression, median (range)			
	18 (5–101)	17.5 (6–115)	
New treatment after synovectomy	N = 117	N = 74	
No	77	34	
Yes *	40	40	0.007
Synovectomy	21	17	
EBR	11	11	
RSO	-	3	
Systemic	6	3	
(Tumor) prosthesis	4	8	
Tumor status at final follow-up ^†^	N = 77	N = 34	
No evidence of disease	53	19	
Alive with disease, watchful waiting	18	10	
Alive with disease, (planned) treatment			
Dead of other disease	5	1	
	1	-	

* Sum of observations can be more than total number of individual patients; EBR—External Beam Radiotherapy; RSO—Radiosynoviorthesis; ^†^ for patients not undergoing a subsequent treatment after a one- or two-stage synovectomy.

## Data Availability

The data presented in this study are available on request from the corresponding author.

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
