# Peer review of "One-Stage Synovectomies Result in Improved Short-Term Outcomes Compared to Two-Stage Synovectomies of Diffuse-Type Tenosynovial Giant Cell Tumor (D-TGCT) of the Knee: A Multicenter, Retrospective, Cohort Study"

_cancers, 2023, doi:10.3390/cancers15030941_

Round 1

Reviewer 1 Report

The Authors aimed to assess whether a single stage double approach synovectomy is superior two a staged approach in case of diffuse GCT.

The topic is extremely interesting. The study well designed and the paper well written.

I only have a few minor concerns.

Please provide a rational for choosing one vs two stage synovectomy. Was the latter chosen in case of larger lumps?

In the two stage group, which was the median time between stages? Did this influence complication rates?

Figure 1: low quality. Please provide more detailed images. Maybe, an anatomic drawing might help more.

Author Response

The Authors aimed to assess whether a single stage double approach synovectomy is superior two a staged approach in case of diffuse GCT.

The topic is extremely interesting. The study well designed and the paper well written.

Thank you for reviewing our manuscript and your kind words

I only have a few minor concerns.

Please provide a rational for choosing one vs two stage synovectomy. Was the latter chosen in case of larger lumps?

There was no clear rational for choosing one vs two-stage synovectomies in the participating centers. The choice was mainly based on the experience of centers or preference of individual surgeons in centers which performed both procedures. However, due to the retrospective design, selection biases may have been introduced, which we also mentioned in our limitations (lines 283-288). Larger lumps were no exact criteria to perform a two-stage synovectomy. Unfortunately, we do not have data regarding the tumor size to quantify this.

In the two stage group, which was the median time between stages? Did this influence complication rates?

The median time between the two interventions was two months, which we mentioned in the results section (lines 182-183). While this may have influenced the rate of short-term postoperative complications such as wound-infection or wound problems, this was also one of the primary outcomes of the study (to assess whether there was a difference in short-term postoperative outcomes). We do not believe this period had a significant influence on longer-term complications.

Figure 1: low quality. Please provide more detailed images. Maybe, an anatomic drawing might help more.

We have provided a higher quality image of figure 1, which can possibly expanded in an external viewer to study the details.

Reviewer 2 Report

Thank you for giving me an opportunity to review your manuscript. This study is interesting but I have a concern. Tumor size is an important factor related to local recurrence rate and postoperative function, but was not examined in this study. If the tumor sizes were significantly different between the two groups, this comparative study would be unfeasible.

Author Response

Thank you for taking the time to review our manuscript.

Although we agree that data regarding the tumor size could add value to this study, we do not believe this would have a great impact on the primary goal of this study: comparing the short-term postoperative outcomes between the two interventions (one- vs two-stage synovectomies).

However, we acknowledge that the tumor size may play a role in relation to local recurrence rates, which was a long-term secondary outcome in this study. Unfortunately, we do not have data regarding the tumor size to analyze this. Therefore, we have added the lack of tumor size as limitation in our manuscript (lines 293-295). On the other hand, we also want to mention that the study of Mastboom et al. 2019 “Surgical outcomes of patients with diffuse-type tenosynovial giant-cell tumors: an international, retrospective, cohort study” also analyzed the tumor size as possible risk factor and found no significant difference in recurrence rates between small and large tumors (<5cm and >5cm).

Round 2

Reviewer 2 Report

The authors replied appropriately and I have no more concerns.